# Preconception Care in a Remote Aboriginal Community Context: What, When and by Whom?

**DOI:** 10.3390/ijerph17103702

**Published:** 2020-05-24

**Authors:** Emma Griffiths, Julia V Marley, David Atkinson

**Affiliations:** 1The Rural Clinical School of Western Australia, The University of Western Australia, Broome, WA 6725, Australia; julia.marley@rcswa.edu.au (J.V.M.); david.atkinson@rcswa.edu.au (D.A.); 2Kimberley Aboriginal Medical Services, Broome, WA 6725, Australia

**Keywords:** preconception, aboriginal health, reproductive health, primary care, preventive care

## Abstract

Preconception care (PCC) is acknowledged as a vital preventive health measure aiming to promote health today and for subsequent generations. We aimed to describe the content and context of PCC delivery in a very remote Australian Aboriginal Community Controlled Health Service setting. A retrospective audit was undertaken to identify what PCC was delivered between 2011 and 2018 to 127 Aboriginal women who had at least one pregnancy during this period. Of 177 confirmed pregnancies, 121 had received PCC prior to the pregnancy. Sexually transmissible infection screening (71%) was the most common care delivered, followed by folic acid prescription (57%) and smoking cessation support (43%). Younger women received PCC less often, particularly screening for modifiable pregnancy risk factors. Rates of prediabetes/diabetes, albuminuria, overweight/obesity and smoking were high amongst those screened (48–60%). PCC was usually patient-initiated and increased significantly over the audit period. Presentation for antenatal care in the first trimester of pregnancy was high at 73%. Opportunities to increase PCC delivery include integration with routine health checks, pregnancy tests and chronic disease programs. PCC programs codesigned with young people are also recommended. All primary care providers should be supported and assisted to provide opportunistic PCC and health promotion.

## 1. Introduction

Preconception care (PCC) is defined as “the provision of biomedical, behavioural and social health interventions to women and couples before conception occurs” [1]. The preconception period provides an important window of opportunity where lifestyle and other changes can have a positive impact on subsequent generations. In recent years, our understanding of the importance of the developmental origins of health and disease has increased, resulting in a paradigm shift from preventive intervention during pregnancy to before conception [2]. The impetus for this change comes from accumulating evidence linking maternal health risk factors in the preconception period to health risk factors in offspring [3,4,5]. The need to support the health of prospective parents is now recognised as an important part of the “life-course” approach to preventive health and health promotion outlined in key international [1], regional [6] and local strategic documents [7].

Despite a broad consensus on the importance of PCC, data from a diverse selection of healthcare settings demonstrates the challenge of achieving good population coverage. In Ethiopia, self-reported utilization of PCC was low (18.2%). The most common component accessed was micronutrient supplementation, with psychological support accessed the least [8]. Utilisation of PCC was positively predicted by health knowledge, coexistence of chronic health problems, partner support and previous adverse pregnancy outcomes. Similar predictors of PCC access were identified in a Chinese cohort [9], where participation in PCC (40% of women) was found to be much less than awareness of PCC (90%). In the United States, the Pregnancy Risk Assessment Monitoring System found that approximately a third of women self-reported access to PCC, with the lowest rates of access in the youngest age group [10].

In Australia, routine provision of PCC is recommended by the Royal Australian College of General Practitioners [11] and the Royal Australian and New Zealand College of Obstetricians and Gynaecologists [2,12]. The advocated standard of care includes, but is not limited to, folic acid supplementation, weight optimisation, medication and chronic disease review, and mental health screening and support for all women planning a pregnancy. Modifiable pregnancy risk factors are common in the Australian population. Results from the Australian Longitudinal Study on Women’s Health have shown that in the three years preconception, 90% of Australian women underconsumed fruit and vegetables, more than a third did not meet physical activity goals, and a fifth were overweight or obese by 23 years of age [13]. Uptake of recommended preconception folate (51%) and iodine (37%) supplements also showed room for improvement [14]. Implementation strategies with a population-based focus (including a broad primary care-based approach) increase the potential for benefit, including for the estimated quarter of pregnancies in Australia that are unplanned [15].

To date, most Australian literature has focussed on population subgroups at high risk by virtue of comorbidity, for example, women with diabetes mellitus. Available data show that Australian women with diabetes received preconception advice less than half the time [16] and report that the majority of PCC discussions were initiated by them, rather than their provider [17]. In contrast, a group of Australian health providers self-reported high rates of adherence with PCC guidelines for women with type 2 diabetes, but described care provision as opportunistic and provider-initiated, with few women presenting specifically for PCC [18].

Aboriginal and Torres Strait Islander women represent a particularly important priority group for PCC delivery due to higher rates of pregnancy risk factors and adverse perinatal outcomes [19]. Rates of diabetes, preterm birth, low birthweight and obesity are more common in Aboriginal and Torres Strait Islander pregnancies than non-Indigenous pregnancies [20]. Aboriginal and Torres Strait Islander women of reproductive age also have higher rates of risk factors such as smoking and risky alcohol consumption [21]. Preconception adiposity and features of metabolic syndrome have been linked with pregnancy risk factors and adverse perinatal outcomes for Aboriginal and Torres Strait Islander women [22,23]. It is critical to address risk factors for perinatal adverse outcomes early, ideally during preconception, given the link to chronic conditions later in life for Aboriginal and Torres Strait Islander people [24].

Whilst suboptimal uptake of PCC has been demonstrated in several countries, these studies have relied on patient self-report. Electronic medical records provide an alternative data source and are less prone to recall bias. This should provide additional insight into PCC provision at a community level in Australia, which is currently lacking. We therefore aimed to describe what components of PCC were delivered by individual pregnancy; the prevalence of key pregnancy risk factors that were identifiable in the preconception period; the type of healthcare worker delivering PCC and how PCC is being delivered over time. This report follows from previous work examining contraceptive use and reproductive decision making in participating communities [25,26].

## 2. Materials and Methods

### 2.1. Study Type

The study design was a retrospective observational cohort study. Patterns of PCC delivery were described using an audit of the electronic medical record system (MMEx; ISA Technologies, Perth, WA, Australia).

### 2.2. Participants

Participants were Aboriginal women with a confirmed pregnancy. Pregnancies were identified by “Antenatal Care Plan” assignment (a searchable electronic medical record system clinical item that provides prompts and recalls) or positive pregnancy test during the audit period. Inclusion and exclusion criteria are depicted in Figure 1.

### 2.3. Person, Place, Time

The setting was three very remote communities in the Western Desert, a remote area of the Kimberley region of Western Australia which has been described elsewhere [25,26]. The audit period was 1 January 2011–1 September 2018. The three participating primary care clinics have an Aboriginal Community Controlled Health Service (ACCHS) model of care [27], that is nurse and Aboriginal health worker led, with one general practitioner (GP) and one midwife covering the three clinics. The midwife position was rotational (six weeks on, three weeks off) and not always backfilled. At the census date (1 September 2018), the same midwife had worked in the community since April 2015.

### 2.4. Data Analysis

Initial full file review was conducted to develop a keyword list that identified all relevant PCC consultations: “concep”, “contracept”, “conceive”, “preg”, “hcg”, “impla”, “folic” or “fola”. These were then used to search consultation notes. Identified consultations were then screened and coded as a “preconception consultation” if they contained any relevant PCC (Appendix A) or if the reason for presentation was to request PCC, a pregnancy test or cessation of contraception. The preconception period for any pregnancy was defined as the time since the previous pregnancy or, in those with no previous pregnancy, since the current electronic medical record system was implemented in early 2010.

For each preconception consultation, we recorded date of consult, date of last delivery (where applicable), estimated due date, clinic of contact, clinical designation of staff member attending, current contraception use, reason for presentation and any components of PCC delivered. “Received PCC pregnancies” were defined as those receiving any key component of PCC (defined as folic acid supplementation, nutrition and weight assessment, smoking cessation, alcohol and illicit substances, chronic diseases, vaccinations, sexually transmissible infections (Appendix A)), and were compared to “Did not receive PCC” pregnancies. Some women had more than one pregnancy in the audit period (Figure 1). In these cases, pregnancy outcomes, risk factors identified prior to pregnancy and PCC were described separately for each pregnancy.

Chronic disease screening results, care plan assignment for diabetes and proteinuria, body mass index (BMI, kg/m^2^) values and smoking status were used to describe screening for and prevalence of preconception risk factors. Prediabetes was defined as glycated haemoglobin (HbA1c) 5.7–6.4%; diabetes was defined as HbA1c ≥ 6.5% mmol/mol or allocated care plan; albuminuria was defined as elevated albumin creatinine ratio (≥3.0 mg/mmol as per Kimberley regional protocol [28]) or allocated care plan. If screening for mental health issues or family and domestic violence was completed, advice regarding normal reproduction and conception given, or cervical cancer screening performed, this was also noted (Appendix A).

Data were compiled in Microsoft Excel 2016 (Microsoft), then imported into Stata 14 (StataCorp). PCC was analysed by pregnancy and by individual consultation to describe overall clinic activity. Results are presented as tables with proportions. Generalized estimating equations, adjusted for correlations within the same woman, for a binomial family using the logit link function, were used to evaluate differences between “Received PCC” and “Did not receive PCC” pregnancies. Postestimation was then used to determine if there was a linear trend over gestation at first antenatal visit (<6, 6–12, 12–24 and 24 weeks and over), age group (15–19, 20–24, 25–29, 30–34 and 35 years and over) and chronic disease screening. Similarly, postestimation following generalized estimating equations, adjusted for correlations within the same woman, for a normal family was used to determine if there was a linear trend for care provision over time. This included number of key components of PCC addressed, number of times key components were addressed per pregnancy and number of consultations at which PCC was delivered. A *p*-value of less than 0.05 was considered significant.

### 2.5. Ethics

This project received ethics approval from the Western Australian Aboriginal Health Ethics Committee (reference 585) and was supported by the Kimberley Aboriginal Health Planning Forum Research Subcommittee.

## 3. Results

### 3.1. Participants and Pregnancy Characteristics

Inclusion criteria were met for 177 pregnancies in 127 Aboriginal women (Figure 1). Median age at pregnancy confirmation was 23.8 years (interquartile range 20.6 to 29.1). In 121 of 177 pregnancies (68%), at least one key component of PCC was delivered (“Received PCC” pregnancies). Of women with multiple included pregnancies (n = 40), one woman had no PCC, 26 had some PCC in all pregnancies, and 13 had some PCC for at least one but not all pregnancies. Of the “Did not receive PCC” pregnancies (n = 56), 19 had visits with opportunities for PCC provision (request for pregnancy test or contraception cessation). There was no difference between care groups in pregnancy outcome or prior parity (Table 1). Pregnancies conceived at the extremes of reproductive age were less likely to have received PCC but this was not statistically significant.

Thirty-four of 177 pregnancies did not have an estimated date of delivery recorded due to early pregnancy loss/ectopic pregnancy (n = 27), termination (n = 1), early pregnancy at the census date (n = 3) or loss to follow up during pregnancy (n = 3). Of the remainder, 73% (105/143) of pregnancies had first antenatal contact in the first trimester, with 27% (38/143) attending prior to six weeks gestation. For pregnancies where preconception advice about normal reproduction and conception was delivered (n = 21), nearly all first antenatal visits were in the first trimester (95%, 20/21) and more than half (57%, 12/21) presented at or before six weeks. Gestation at first visit was not independently associated with age or parity (Table 1).

### 3.2. Pregnancy Risk Factors

Rates of screening for chronic disease and risk factors were low with no identified differences between care groups (Table 1). Where screening results were available, rates of prediabetes or diabetes (48%, 38/79), proteinuria (43%, 43/101), BMI above the ideal range of 18.5–25 (42%, 33/79,) and current smoking (60%, 45/75,) were high. Younger women were less likely to have been screened for chronic diseases (linear trend across age groups *p* < 0.01): in 15–19 year olds, 20% (7/35) and 31% (11/35) had a diabetes or proteinuria screening result, respectively, compared to 72% (13/18) and 94% (17/18), respectively, in the over 35 age group. Preconception risk reduction advice was recorded for 42% (19/45) of pregnancies where smoking was identified and 47% (16/34) of pregnancies with a high BMI. Preconception chronic disease management was recorded for 23% (14/61) of pregnancies complicated by prediabetes, diabetes or proteinuria.

### 3.3. Characteristics of Preconception Care Delivered

Five hundred seventy-nine consultations were identified in “Received PCC” pregnancies. Two-thirds were patient-initiated (requests for PCC, pregnancy tests or contraception cessation). A fifth of care was delivered opportunistically with unrelated presentations, and a small percentage (5%, 29/579) with chronic disease management and scheduled health checks (Table 2). PCC was delivered at 28% (45/163) of presentations where women requested pregnancy tests that were subsequently negative. Most care delivery involved nursing staff or Aboriginal health workers, either as sole provider (59%, 343/579) or in conjunction with a GP or GP registrar (19%, 111/579). PCC was delivered at a median of three (interquartile range 2–6) consultations per pregnancy.

A summary of PCC is provided in Table 2. Additionally, advice regarding normal reproduction and conception was delivered prior to 21 pregnancies; family and domestic violence and/or mental health screening was done prior to 21 pregnancies; and 40 pap smears were performed. Investigation for subfertility was delivered eight times; assisted fertility treatment once. PCC delivery showed a small but significant increase over the audit period (Table 3). When removal of contraceptive implant was requested (n = 176 consultations), it was removed in the same consultation 36% of the time (63/176), deferred for PCC in 26% (46/176) of consultations and deferred due to lack of clinic availability or other reasons in 38% (67/176) of consultations. A median of two and maximum of ten consultations were recorded from first request to device removal.

## 4. Discussion

This study comprehensively describes PCC delivery at a community level using primary care electronic medical records to achieve an in-depth analysis. In a clinical area of emerging public health significance, this provides a necessary foundation on which continuous quality improvements can be built. Screening for sexually transmissible infections (STI) and folic acid supplementation were delivered more commonly than health promotion including risk reduction around alcohol and tobacco exposure, diet and physical activity. PCC delivery was most commonly patient-initiated and increased over the audit period. Modifiable pregnancy risk factors were common, underscoring the importance of PCC in providing a head start to a healthy pregnancy.

Our results provide insight into PCC delivery in a remote Australian ACCHS primary care setting. Where PCC was delivered, the most common reasons for presentation were requests for a pregnancy test or to cease contraception. We note that, unlike most regions of Australia, there is no commercial outlet for pregnancy tests. This is therefore a reason for presentation particular to this setting. Similarly, the etonogestrel implant was the most common contraceptive used by women in these communities [25], requiring health provider assistance for removal. PCC was delivered at only 28% of consults where a requested pregnancy test returned a negative result. It took a median of two consultations for contraceptive implant removal; however, for some women it took several more. Reasons for delay included PCC delivery (24%) and unavailability of staff trained in implant removal. Pregnancy testing and contraceptive counselling provide important and potentially underutilised opportunities to provide PCC; however, care must be taken to preserve patient autonomy and engagement. Pressure from a provider to commence or continue contraception can compromise patient trust [29]. Therefore, it is important that timely PCC is delivered when discontinuation is requested. Most PCC was patient-initiated. In qualitative studies, Australian women have expressed support for GPs to be more proactive in raising the topic of PCC [30]. This can occur as part of holistic reproductive health counselling at the time of contraceptive discussions, consistent with the “reproductive life plan” discussion recommended by the Royal Australian College of General Practitioners [11].

In our study and as per the ACCHS model of care more generally, women were most likely to initially receive care from an Aboriginal health worker or nurse. Therefore, these staff members require appropriate training and resources to proactively raise PCC with their patients. The comprehensive primary healthcare model employed by ACCHSs has been shown to increase Aboriginal access to preventive activities such as cervical cancer screening, chronic disease programs, mental health and sexual health [31]. Support for all clinic staff to confidently deliver PCC and integration of PCC into clinic flow should be based on the success of other models that prioritise cultural security and an integrated team approach [32].

Antenatal care was accessed in the first twelve weeks of most pregnancies in this study (73%). This is an encouraging sign of engagement between community and clinic. It is above the 68.6% for the general Australian population [33] and significantly higher than the 58.2% described elsewhere for Aboriginal and/or Torres Strait Islander women living in very remote areas [34]. First trimester antenatal care access is a key measure reported in the Aboriginal and Torres Strait Islander Health Performance Framework [19]. Contributing factors may have included staff continuity: one midwife delivered antenatal care for the latter half of the study period, the benefits of which have been well described elsewhere [35].

Additionally, in almost all of the 21 pregnancies where advice about normal reproduction and conception was delivered, first presentation was in early pregnancy. Although a small group, this highlights a potentially important role of the remote clinic in supporting reproductive health literacy not currently emphasised in PCC guidelines [12]. The importance of this role has been highlighted previously in the course of this project [25], as well as in qualitative studies with young Aboriginal women [36]. Health literacy encompasses the access, understanding, appraisal and application of health information within complex health environments [37], and interventions that enhance health literacy improve adherence to medical treatment [38]. Lower PCC knowledge has been found in minority [39] or socially disadvantaged groups [40,41], in association with low health literacy [42]. Strategies to support reproductive health literacy amongst vulnerable groups are therefore important from a position of equity. Attempts to increase PCC knowledge have shown positive outcomes [43] and materials developed in consultation with the target audience demonstrated improved results [44]. It is important that health literacy is considered both in the design of future interventional studies and as an important outcome in and of itself.

In the Australian primary care setting, time constraints and competing priorities for preventive activities have been described as a barrier to PCC delivery [45], highlighting the need to integrate PCC into existing clinical flow, billable activities (through the Australian Medicare Benefits Schedule) and organisational structures. Increased uptake of routine annual health assessments is a central goal in the Implementation Plan for the National Aboriginal and Torres Strait Islander Health Plan 2013–2023 [46]; however, they were underutilised for PCC in this study. Chronic disease management programs also represent important opportunities for PCC integration. Smoking cessation, weight loss and improved diabetes control are often part of a GP management plan and have direct relevance to fertility and pregnancy outcomes [47]. If chronic disease care and preventive health activities are not integrated with PCC, important opportunities for brief intervention and motivational interviewing will be missed.

We identified two areas that may require specific focus. Firstly, pregnancies conceived at the extremes of reproductive age were less likely to have received PCC. Although this was based on small numbers and not statistically significant, it correlates with US findings that younger women received PCC less frequently [10]. Possible explanatory factors include a lower uptake of preventive health activities in general [46]; a greater likelihood of unplanned, although not necessarily unwanted pregnancy [48]; and health provider reluctance to endorse pregnancy planning for younger persons [49]. Despite a recognition of need, research on what strategies most effectively increase health service access for Aboriginal young people is lacking. Suggestions collated from young people elsewhere in the region [50] should inform codesign and evaluation of young people only or evening clinics and group health promotion sessions. Older women might believe themselves not “at risk” of pregnancy and underutilise contraception [51], and this may also require a targeted approach. Secondly, health promotion was delivered less often than screening for infections and provision of folic acid. Health promotion may be considered more difficult, time consuming or lower on the priority list in a busy clinical environment. Obesity was under-recognised as an indication for PCC in another Australian primary care setting, where weight and mental health were the least common components of PCC delivered [52].

In parallel with this study, a regional Preconception Care Protocol was developed through the Kimberley Aboriginal Health Planning Forum [53] and supporting resources have been drafted. Other possible approaches include electronic medical record system templates and augmentation of clinic flow (e.g., PCC resources stored with pregnancy tests or STI treatment kits). High staff turnover in remote areas will likely remain an ongoing challenge for service delivery. Therefore, systems need to be robust, developed in partnership with services and evaluated over time.

The strengths of this study are the comprehensive primary care-based approach and the inclusion of all pregnancies, not just high-risk subgroups. The length of the audit period is sufficient to capture activity over time, and the inclusion of all care providers gives a “whole of clinic” perspective. It is limited by the retrospective design and dependence on provider documentation, which can underestimate activity. Services with incidental preconception benefit were not included (e.g., weight loss advice where pregnancy intentions or outcomes were not discussed or recorded). We therefore described only PCC delivered by intent, which does not represent the total preventive health activities delivered by the service. PCC delivered to women who did not become pregnant is not captured: women who had difficulty conceiving are therefore under-represented. Consultation notes that did not contain words that met our search criteria were not reviewed; therefore, we are unable to quantify the potential for provider-initiated, opportunistic PCC at the time of presentation for unrelated health concerns. As a regional study, the findings may not be generalisable to other locations and models of care.

## 5. Conclusions

Overall, our results suggest good engagement between the health service and Aboriginal women, with most presenting early for antenatal care. PCC delivery increased over time but needs to be further supported and appropriately resourced as a priority to reduce the transgenerational impact of health risk factors and secure the best possible health for future generations. This will be best achieved through integration with routine health checks, pregnancy tests and chronic disease programs, and through upskilling and support of care providers within a comprehensive primary care context.

## Figures and Tables

**Figure 1 ijerph-17-03702-f001:**
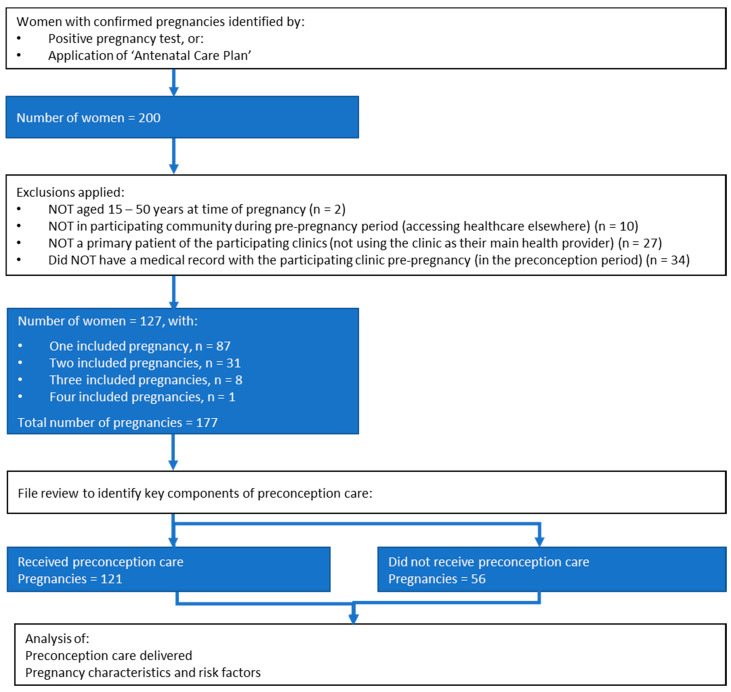
Search strategy applied to identify women with confirmed pregnancies and the preconception care they received (audit period 1 January 2011–1 September 2018).

**Table 1 ijerph-17-03702-t001:** Characteristics of confirmed pregnancies, by preconception care (PCC) group (n = 177).

	n	Received PCC	Did Not Receive PCC	Logistic Regression	*p*
		n (%)	n (%)	OR (95% CI)	
Age at first antenatal visit:					
15–19 years	35	22 (18)	13 (23)	0.66 (0.27–1.60)	0.35
20–24 years	68	49 (41)	19 (34)	1 ^λ^	
25–29 years	33	24 (20)	9 (16)	1.03 (0.39–2.73)	0.95
30–34 years	23	17 (14)	6 (11)	1.10 (0.38–3.21)	0.86
35 years and over	18	9 (7)	9 (16)	0.39 (0.11–1.41)	0.15
Pregnancy outcome:					
Live birth	141	95 (78)	46 (82)	1 ^λ^	
Miscarriage	28	19 (16)	9 (16)	1.02 (0.32–3.24)	0.97
Other *	8	7 (6)	1 (2)	3.39 (0.40–28.98)	0.27
Gestation at first antenatal visit:					
Unknown ^#^	34	25 (20)	9 (16)	1.27 (0.39–4.09)	0.69
<6 weeks	38	30 (25)	8 (14)	1.71 (0.66–4.44)	0.27
6–12 weeks	67	46 (38)	21 (38)	1 ^λ^	
12–24 weeks	26	13 (11)	13 (23)	0.46 (0.18–1.17)	0.10
24–36 weeks	12	7 (6)	5 (9)	0.64 (0.19–2.18)	0.48
Parity prior:					
0	56	39 (32)	17 (30)	1 ^λ^	
1	53	38 (32)	15 (27)	1.10 (0.49–2.48)	0.81
2	30	21 (17)	9 (16)	1.02 (0.41–2.54)	0.97
3	18	13 (11)	5 (9)	1.13 (0.35–3.64)	0.83
4 and greater	20	10 (8)	10 (18)	0.44 (0.13–1.48)	0.18
Diabetes status preconception (by HbA1c (mmol/mol) or care plan allocated):					
Unknown (no result)	98	64 (53)	34 (61)	n.a.	
Normal (<5.7%)	41	30 (25)	11 (19)	1 ^λ^	
Prediabetes (5.7–6.4%)	13	11 (9)	2 (4)	2.02 (0.37–10.86)	0.41
Diabetes (≥6.5%)	25	16 (13)	9 (16)	0.65 (0.21–2.00)	0.45
Albuminuria status preconception (by albumin: creatinine ratio (mg/mmol) or care plan allocated):					
Unknown (no result)	76	53 (44)	23 (41)	n.a.	
Normal (<3.0)	58	38 (31)	20 (36)	1 ^λ^	
Elevated (≥3.0)	43	30 (25)	13 (23)	1.21 (0.52–2.81)	0.65
BMI preconception (kg/m^2^):					
Unknown (no result)	97	61 (50)	36 (65)	n.a.	
<18.5	20	13 (11)	7 (12)	0.44 (0.12–1.65)	0.23
18.5–25	26	21 (17)	5 (9)	1 ^λ^	
25–30	18	13 (11)	5 (9)	0.62 (0.13–2.93)	0.55
>30	16	13 (11)	3 (5)	1.03 (0.21–5.14)	0.97
Smoking behaviour preconception:					
Unknown (no record)	102	67 (55)	35 (63)	n.a.	
Current smoker	45	35 (29)	10 (18)	1 ^λ^	
Ex-smoker	10	6 (5)	4 (7)	0.43 (0.11–1.60)	0.21
Never smoked	20	13 (11)	7 (12)	0.53 (0.13–2.13)	0.37
TOTAL		121 (100)	56 (100)		

* Includes termination of pregnancy (n = 1), ectopic pregnancy (n = 2), and pregnancies ongoing at census date (n = 5). ^#^ Includes termination of pregnancy (n = 1), miscarriages (n = 25), ectopic pregnancy (n = 2) or pregnancies lost to follow up or where no estimated delivery date was recorded (n = 6). ^λ^ Largest group used as reference group for logistic regression (generalized estimating equations, adjusted for correlations within the same woman, for a binomial family using the logit link function). n.a. = not applicable.

**Table 2 ijerph-17-03702-t002:** Characteristics of preconception care delivered.

Reason for Presentation, by Consultation (n = 579):	n (%)
Requesting pregnancy test	163 (28)
Requesting cessation of contraception	135 (23)
Unwell or other unrelated health concern	114 (20)
Preconception care	70 (12)
Sexual health	30 (5)
Chronic disease management or scheduled health check	29 (5)
Multiple reasons	21 (4)
Requesting check-up	17 (3)
Designation of staff member/s, by consultation (n = 579):	
Nurse or Aboriginal health worker only	343 (59)
General practitioner * and nurse or Aboriginal health worker	111 (19)
General practitioner * only	98 (17)
Midwife	22 (4)
Other	5 (1)
Components of care delivered, by pregnancy (n = 121) **:	
Sexually transmissible infections	86 (71)
Folic acid	69 (57)
Smoking cessation and avoidance	52 (43)
Nutrition and weight	44 (36)
Alcohol and illicit substances	31 (26)
Chronic disease management	20 (17)
Vaccinations	14 (12)

* General practitioner includes registrars (doctors completing general practice training). ** Number (%) of pregnancies receiving care, does not sum to 100%.

**Table 3 ijerph-17-03702-t003:** Preconception care (PCC) over time.

Year	Pregnancies, Total (n)	“Received PCC” Pregnancies (n (%))	Of Those Receiving Care: Components of PCC Delivered, by Pregnancy (Median)		
			Number of Key Components **	Number of Times ***	Number of Consultations
2011	13	8 (53)	2.5	3	2.5
2012	29	18 (58)	1	1	3
2013	26	17 (52)	3	3	3
2014	22	13 (57)	2	3	2
2015	37	24 (60)	3	4	4
2016	18	14 (74)	3	4	3.5
2017	23	18 (70)	3.5	5.5	5.5
2018	11 *	9 (75)	4	7	5
Total	177	121 (100)			
*p* ^#^			0.105	0.044	0.003

* Incomplete year, inclusive of 1 January 2018–1 September 1 2018, excluded from trend analysis. ** Key components (n = 7): folic acid supplementation, nutrition and weight assessment, smoking cessation, alcohol and illicit substances, chronic diseases, vaccinations, sexually transmissible infections. *** Total includes multiple counts if topics were addressed more than once, e.g., repeat folic acid provision was counted twice. ^#^ Postestimation following generalized estimating equations, adjusted for correlations within the same woman, for a normal family was used to determine if there was a linear trend for care provision over time.

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
