# Peer review of "Preconception Care in a Remote Aboriginal Community Context: What, When and by Whom?"

_ijerph, 2020, doi:10.3390/ijerph17103702_

Round 1

Reviewer 1 Report

  • Abstract: Needs revision and should be re-written. Most of the sentences seem incomplete, and not articulated.
  • Keywords: ‘Aboriginal community’ instead of ‘Aboriginal health’ would be more appropriate.
  • Introduction:
    • The definition of PCC should come first, and the definition needs re-checking.
    • Page 1, Line 31: Improving the health of prospective parents has attracted increasing attention: English needs to be
    • The overall ‘Introduction’ part needs re-writing. Sentence articulation and English is very poor.
  • Materials and Methods:
    • Figure 1: First box: Application of ‘Antenatal Care’ care plan; it should be ‘Antenatal Care’ plan. The figure is not very clearly presented, and difficult to understand.
    • Exclusion criteria: difficult to understand
  • Discussion:
    • Please use reported tense because situations change and so is information and people's practices.
    • Recommendations and conclusions should be written with more assertive language to be able to influence policy.
  • The authors need to revisit abstract after addressing all comments in the manuscript.
  • Extensive editing of English language and editing is required.

Reviewer 2 Report

This is a very well written article on an important topic. I only have small comments for revision:

Introduction

  • Line 59 – the standard of care is more comprehensive than this. I appreciate it is not feasible to list them all here. Suggest acknowledging there are other components.
  • Line 81. This is important and would benefit from being reworded. Suggest two clearer sentences here. Suggest “Whilst there is an existing data set describing suboptimal uptake of PCC in several countries, these studies have utilized patient self-report rather than electronic medical records and insight into PCC 83 provision at a community level in Australia is lacking.

Materials and Methods

Please clearly define the study type, retrospective observational cohort study.

Recommend this section be divided into subheadings.

  1. Study Type
  2. Participants
    1. Person, place, time
    2. Inclusion & Exclusion criteria
  3. Data Analysis
  4. Ethics

  • Line 92: Suggest a reference provided to the relevant section of the NACCHO website to support the statement of an ACCHO Model of care.
  • Line 97: Please define what an antenatal care plan assignment is? Is this a searchable item in the eMR (As this is an international journal not all will be familiar with Australian data systems).
  • Line 124: HbA1c units of measurement – The % units of measurement for HbA1c cut-offs are more commonly used. Suggest changing to these units of measurement (they are listed in the referenced guideline)

Figure 1

In the first box please make it clearer that only women with a confirmed pregnancy were included for analysis

Audit time frame is in the Figure 1 title, it can be removed from the 1st box.

Results

  • Line 145: the proportion of women who did not receive PCC should be 32% (56/177)
  • Line 154: Please specify that this is in women who had received PCC (it just makes it easier for the reader to digest your results)
  • Line 158-161: where are these 21 women from and where did they receive this information?
  • Line 174: Is this risk reduction advice for smoking in the preconception period rather than the pregnancy period?

Table 1

Please check your data and percentages (there are multiple areas that need correction – all are not listed here)

  • In the group that received PCC by age at first antenatal visit, the percentages add up to 99%. 20-24 years should be 41%.
  • In the group that received PCC care by pregnancy outcome the cumulative percentage is 101%
  • Again in the group that received PCC by gestation at first antenatal visit the cumulative percentage is 101%
  • In the group that did not receive PCC by Diabetes status pre pregnancy the cumulative percentage is 101%

Discussion

Lines 248 – 261: It is not clear to me who this group of women is nor the advice they received. While health literacy is important it does not fit here.

Otherwise a great discussion.

Appendix A

Cervical Screening is not included in the PCC audit tool. Suggest this is included or the authors provide justification for this.

Round 2

Reviewer 1 Report

Thanks for addressing the comments and revising the paper based on the comments and feedback.